# Convergent genomic and molecular features predict risk of metachronous metastasis in clear cell renal cell carcinoma

Marjan M. Naeini [1,2,3,5] ✉, Mengyuan Pang[1,5], Neha Rohatgi[1], Sinem Kadioglu[1], Umesh Ghoshdastider[1], Renzo G. DiNatale[4], Roy Mano[4], A. Ari Hakimi [4] & Anders Jacobsen Skanderup[1] ✉

## Abstract

**Background** The molecular features determining the risk of metachronous metastases in clear cell renal cell carcinoma (ccRCC) are poorly defined. This study aimed to identify molecular factors associated with the risk of metachronous metastasis.

**Methods** Using a systematic tumor transcriptome deconvolution approach, we investigated the genomic and transcriptomic profiles of 192 ccRCC primary tumors with extended clinical follow-up to identify cancer- and stromal cell-specific molecular features associated with metastatic risk. Based on these features, we applied multivariate Cox regression to develop a compact 5-gene predictive model for metachronous metastasis.

**Results** At the genomic level, we identify a significantly higher frequency of copy number loss at 1p31-36 in primary tumors that later progress with metastases. Tumor transcriptome deconvolution identifies significant down-regulation of epithelial cell polarity, including *PATJ* (1p31), and fatty acid metabolism, including *CYP4A11* (1p33), in cancer cells of tumors that develop metastatic progression. We develop and benchmark a compact 5-gene predictive model (5G) that demonstrates improved accuracy over existing ccRCC gene signatures in the prediction of metachronous metastasis risk.

**Conclusions** Overall, our study highlights convergent genomic and transcriptomic alterations in chromosome 1p, driving dysregulation of epithelial cell polarity and fatty acid metabolism, as putative risk factors of metachronous metastasis in ccRCC.

## Plain Language Summary

Clear cell renal cell carcinoma (ccRCC) is a common type of kidney cancer. In around one in three people with ccRCC, the cancer spreads to other parts of the body 100 days or later after surgery. This condition, known as metachronous metastasis (MM), is often difficult to treat. It is unclear which patients are more likely to develop MM. In this study, we analyzed tumor samples from 192 patients with ccRCC to identify molecular changes linked to development of MM. We found that specific changes on chromosome 1 in the tumor's DNA and RNA were strongly associated with MM. Using these findings, we developed a mathematical model to predict the risk of MM. Our findings may help identify people at high-risk of MM and also guide development of future treatments.

Metastatic clear cell renal carcinoma (ccRCC) has a 5-year survival rate of ~10% and is generally considered incurable[1]. Metastatic spread of ccRCC is often diagnosed during pre-operative assessment (i.e., synchronous spread). However, around one-third of ccRCC patients with localized disease eventually relapse with metastatic progression following curative surgery[2]. The third prespecified interim analysis of the KEYNOTE-564 trial demonstrated that adjuvant pembrolizumab significantly and clinically meaningfully improved overall survival compared to placebo in participants with ccRCC at high risk of recurrence after surgery[3]. The ability to identify

ccRCC patients with an increased risk of developing metachronous metastasis could lead to improved treatment strategies, such as active surveillance and additional adjuvant therapy using pembrolizumab[4,5].

Previous studies have established the genomic and molecular landscape of ccRCC[6–8] and demonstrated widespread metabolic reprogramming[9], immune infiltration signatures[10], as well as recurrent alterations of the PI3K/AKT/mTOR pathway[6,7,11]. Recurrent copy number alterations (CNA) include loss of chromosome arms 1p, 3p, 4q, 6q, 8p, 9p and gains of 1q, 2q, 5q, 7q, 8q, 12p, and 20q[7,8,11,12]. In the metastatic setting,

[1]Genome Institute of Singapore (GIS), Agency for Science, Technology and Research (A*STAR), 60 Biopolis Street, Genome, Singapore, Republic of Singapore. [2]Garvan Institute of Medical Research, Sydney, NSW, Australia. [3]Faculty of Medicine, St Vincent's Clinical School, University of New South Wales, Sydney, NSW, Australia. [4]Dept. of Surgery, Urology Service, Memorial Sloan Kettering Cancer Center, New York, NY, USA. [5]These authors contributed equally: Marjan M. Naeini, Mengyuan Pang. ✉e-mail: m.naeini@garvan.org.au; skanderupamj@a-star.edu.sg

genomic loss of 9p and 14q have been reported as clonally selected and expanded driver events within metastatic lesions[8]. Patients with metastatic disease and low body mass index (BMI) tend to have a poor prognosis, and studies have suggested a role for dysregulated fatty acid metabolism in the metastatic progression of ccRCC[13].

Efforts to decipher the transcriptomic features of metastatic ccRCC tumors have led to the development of putative prognostic biomarkers, including the ClearCode34 risk predictor for localized ccRCC[14] a 16-gene recurrence score[15], and an RNA-based 31-gene cell cycle progression (CCP) signature[16]. Furthermore, metastatic progression is often accompanied by marked changes in the tumor microenvironment (TME)[17]. A recent study identified intratumoral myeloid inflammation as a key driver of metastasis in ccRCC[18]. Moreover, Alchahin et al. investigated single cell gene expression data from tumors diagnosed with synchronous metastasis and proposed biomarkers associated with synchronous metastasis[19]. These studies underline the influence of the TME on the metastatic progression of ccRCC. However, our understanding of cancer and stroma cell features rendering patients with localized tumors at risk of progressing with metachronous metastasis remains nascent.

Here we sought to identify cancer and stromal cell molecular characteristics of primary ccRCC tumors associated with the development of metachronous metastasis. We conducted an integrative genome and transcriptome analysis of 192 primary tumors with prolonged clinical follow-up (up to 8 years) to accurately distinguish indolent tumors from those that subsequently developed metastasis. We comprehensively investigated potential genomic features associated with the development of metachronous metastasis. We then performed a deep comparative analysis of the tumor transcriptomes in these patient subsets, comparing both inferred immune cell infiltrates as well as deconvoluted cancer and stromal-cell tumor transcriptomes. Finally, we developed and benchmarked a metachronous metastasis risk model based on the top predictive gene expression features in the dataset.

Overall, our study provides a systematic analysis of the molecular features associated with metachronous metastasis in ccRCC. We observe that metastasis progression is linked to copy number loss at chromosome 1p31-36 and to down-regulation of epithelial cell polarity and fatty acid metabolism in cancer cells, including *PATJ* (1p31) and *CYP4A11* (1p33). These features represent candidate biomarkers for improved risk stratification of primary tumors at risk of metastatic progression. And our 5-gene model based on the identified significant features can help improve the patient risk prediction of metastasis.

## Methods
### Patient cohorts
We obtained data for 537 primary ccRCC tumor samples through The Cancer Genome Atlas (TCGA). Level 3 RNA-seq profiles (RNAseqV2 normalized RSEM) were downloaded from the UCSC Xena browser (RRID:SCR_018938). We obtained Mutation Annotation files (MAF), level 3 Copy number variation and segmentation (CNA) from the Broad Institute GDAC Firehose website (RRID: SCR_026267). All data mentioned above are publicly available and data URLs are provided in Supplementary Table 1.

We obtained curated clinical data with additional follow-up annotation of metastatic disease from the clinical TCGA (cTCGA) consortium. The median follow-up period was 23 months, with a maximum follow-up of 8 years. Cohorts were then stratified by median follow-up and patients' metastatic status at the time of nephrectomy. Patients who developed metastasis within 100 days of surgery were categorized as synchronous metastatic cohort (SM, $n = 80$). Patients with diagnosed metastatic disease after 100 days and up to 8 years were grouped as metachronous metastatic cohort (MM, $n = 44$). Finally, patients that did not develop metastasis within their follow-up period were classified as indolent cohort (IN, $n = 68$). The median follow-up for IN cohort was 51 months (Supplementary Data 2).

As a validation cohort, we used data from the Sato et al.[6]. This dataset contained bulk tumor gene expression data (Agilent microarray), which is publicly available from ArrayExpress (RRID:SCR_002964). These samples

($n = 101$) were stratified according to the TCGA cohort, yielding an SM ($n = 32$) and IN ($n = 69$) cohort (Supplementary Fig. 3). The median follow-up for the IN cohort was 51 months, equivalent to the TCGA IN cohort. We mapped the probe identifiers to HGNC gene symbols and applied arithmetic mean for single HGNC symbols with multiple mapped probe identifiers.

Specific IRB approval was not obtained for this study as extended follow-up data collection for metastatic recurrence was conducted under the original TCGA framework and IRB approvals.

### Mutation and CNV analysis
Two-sided Fisher's exact test was implemented to compare gene mutation frequencies of MM and SM tumors with IN tumors. Copy number alteration differences between cohorts were examined using a non-parametric two-tailed Wilcoxon rank-sum test. A two-tailed Wilcoxon rank-sum test was performed to compare bulk tumor expression of the genes located in 1p and 19q between MM and IN cohorts. Pearson correlation was used to examine correlations between CNV and bulk tumor expression in primary tumors, IN and MM samples.

### Tumor Purity Estimation
Tumor purity and intratumor heterogeneity were estimated using the approach described in Ghoshdastider et al.[20]. Estimates of tumor purity for TCGA cohort are based on a consensus of four different methods (PurBayes[21], ESTIMATE[22], ASCAT[23], and AbsCN-seq[24]), using gene expression profiles, CNA segmentation, and SNV variant allele frequency (VAF) data of individuals. SATO purity estimates were inferred using PurBayes and ESTIMATE.

### Tumor gene expression deconvolution
We decomposed bulk tumor expression profiles to stroma and cancer cells expression using the TUMERIC model[20]. Bulk tumor expression values were log-transformed, $\log_2(X + 1)$, before deconvolution. Genes with median expression (FPKM) < 2 were excluded from this analysis.

### Differential gene expression analysis
We used pair-wise class comparisons to identify genes differentially expressed between IN patients and the other two cohorts. To identify genes with altered expression restricted to a specific compartment (cancer vs. stroma), we developed a Differential Expression (DE) score. Log2 transformed expression values ($e$) inferred from TUMERIC for cancer ($c$) and stroma ($s$) were obtained. We then calculated the extent that a gene showed gene expression change (comparing groups 1 and 2) primarily in cancer or stroma compartment with the following DE score:

$$DE = log_2 \frac{abs(e_{c2}/e_{c1})}{abs(e_{s2}/e_{s1})}$$

Here positive DE scores denote genes with higher expression change in the cancer compartment, and negative scores denote genes with higher DE in the stroma. A gene-wise two-tailed permutation test (non-parametric randomization test) was implemented to examine the statistical significance of the DE score. In this test, we estimated the *NULL* distribution of the DE statistic using random shuffles of the sample-to-group assignments. In each shuffle/permutation, cancer/stroma expression was estimated by TUMERIC and DE scores were computed. The $p$-values were evaluated based on n = 50,000 permutations and computed using the fraction of *NULL* DE scores that are at least as extreme as the observed DE score for a given gene. The $p$-values were corrected for multiple testing using the Benjamini-Hochberg method. A similar method was performed for DE analysis in the SATO et al cohort.

### Cell Type Enrichment Analysis
Cell type deconvolution of cell types in the TME was performed using bulk tumor RNA-seq using the xCell R package[25], Consensus[TME][26], and

Cibersort[27]. xCell is a gene signature-based enrichment method that decomposes the expression data into 64 different cell types. Consensus[TME] combines gene set-based methods and regression-based methods using multiple statistical tests to infer cell type enrichments. Cibersort is a deconvolution algorithm that estimates the abundances of 22 immune cell types using previous knowledge of immune cell expression signatures. To facilitate the analysis, Cibersort immune cells were grouped together as B cells (B Cells Naïve, B Cells Memory), T-Cell CD4 memory (T-Cell CD4 resting, T-Cell CD4 activated), T-Cell CD4 (T-Cell CD4 Naïve, T-Cell CD4 Memory), Natural Killer cells (NK Resting, NK Activated), macrophages (macrophage M0, macrophage M1, macrophage M2) as one group as well as separate groups, Dendritic cells (DC Activated, DC Resting), Mast cells (Mast cell resting, Mast cell activated), Monocytes, Plasma cells, Tgd cells and T regulatory cells (Tregs). Tumor samples with a Cibersort deconvolution performance p-value > 0.05 were excluded. Further analysis was performed on the shared cell types of the three methods with cell type fraction >10% in at least in 5 patients. To establish a cell type-wise differential analysis, we compared the median cell proportions of MM and IN patients. The significance level of these comparisons was estimated using a non-parametric Wilcoxon rank-sum test. The *p*-values were adjusted to FDR *q*-values using the Benjamini-Hochberg method.

### Pathway Enrichment Analysis

To investigate differentially expressed genes in cancer and stroma cells, we used the KEGG pathway database (downloaded on September 2018, Supplementary Table 1, RRID:SCR_018145) and pathway enrichment analysis[28]. The Bioconductor (RRID:SCR_006442) annotation package 'org.Hs.eg.db' was used for mapping gene symbols to ENTREZ gene identifiers. The background gene list was defined as shared genes between KEGG pathways and the study datasets ($n = 4505$). To perform enrichment analysis in cancer and stroma discretely, two input gene lists were provided. These gene lists comprised the shared genes between the KEGG pathways and the significant genes in cancer ($n = 679$) and stroma ($n = 720$) gene expression analysis, as shown in Fig. 6 (cancer and stroma p-values < 0.05). A one-sided Fisher's exact test was applied to evaluate the significance level of pathway enrichment. *P*-values were adjusted for multiple testing using the Benjamini-Hochberg method.

### Tight junction and polarity complex analysis

To explore tight junction and polarity complexes, a gene list was derived from the tight junction pathway of the KEGG database. The 120 inferred genes were further curated and annotated as follows: Crumbs complex, Scribble complex, Par complex, Actin assembly, and others. Crumbs, Scribble, and Par complexes are the most well-known complexes responsible for cell polarity. Actin addresses the genes that participate in the structures and dynamics of the actin-based cytoskeleton. All other classifications are defined as 'other regulators'.

### Metabolomic analysis

The Ensembl gene IDs in the RNA-seq data set were mapped to the Entrez IDs used in RECON3 using the BioMart R tool[29,30]. RECON3 was manually checked and curated using information from literature, as previously described[31]. Furthermore, deconvoluted cancer and stromal cell expression for MM and IN tumors was integrated with the RECON3 based on the gene–protein–reaction (GPR) information using the COBRA toolbox[32]. To estimate reaction expression from the deconvoluted cancer cell expression, we evaluated the GPR rule of each reaction while replacing the "OR" and "AND" operators with "max" and "min", respectively[31,33,34].

### Cox regression model predicting metachronous metastasis

To identify biomarkers associated with the risk of developing metachronous metastasis, we performed univariate Cox analyses using 'coxph' function from R package 'survival' on features ($n = 22$) associated with metachronous metastasis in this study. Time to metachronous metastasis after surgery was considered as the endpoint, and only patients with >= 100 days follow-up

and available RNA-seq data were included. 94 out of 112 MM and IN patients met the inclusion criteria. Among them, 41 TCGA patients (median follow-up of 51.22 months) with MM were considered as events, while 53 TCGA patients with IN (median follow-up of 55.56 months) were treated as controls (right censored). Records with missing values in the relevant columns were excluded. Numeric features were analyzed as continuous variables without dichotomization and scaled to enable a fair comparison of coefficients, while the categorical features were analyzed using one-hot encoding. From this analysis, we selected the five most significant gene expressions (*FKBP15*, *SLC31A1*, *CPT2*, *PATJ* and *CALR*) as our signatures to predict metachronous metastasis. Using cross-validation, we then evaluated the predictive power and generalization performance of the identified 5G signature. We stratified samples into 5 folds, fitting multivariate Cox regression on 4 folds and evaluating the model performance on the remaining fold. We estimated matrices including the Akaike information criterion (AIC), the area under the curve (AUC), the concordance index (c-index), likelihood-ratio p-value, specificity and sensitivity to evaluate and compare the models. We repeated the analyses described above 100 times and calculated the mean value of all metrics. We compared the performance of the 5G signature for predicting metachronous metastasis with gene signatures provided by previous studies.[14–16,18,19]

### Log-rank survival analysis

We assessed whether the 5G score is associated with disease-free survival. Two patients without patient vital status information were excluded, leaving 92 patients with MM or IN for analysis with a median follow-up of 53.62 months. Among them, 49 patients experienced metastasis or death. Here we stratified patients into three groups based on their 5G scores. We performed a log-rank test using disease-free survival months and status for the three groups.

### Statistics and Reproducibility

All statistical analyses were conducted in R (v 4.1.3), unless otherwise indicated. Two-sided Fisher's exact tests were used to compare mutation frequencies between cohorts. And two-tailed Wilcoxon rank-sum tests were used to compare CNA, bulk tumor gene expression, and cell type proportions between cohorts. Pearson correlations were applied to evaluate associations between CNV and bulk turmor gene expression. DE in cancer and stroma compartments was assessed using TUMERIC (v1.0). Empirical p-values were derived using two-sided non-parametric permutation test. Significance level of Pathway enrichment was evaluated using one-sided Fisher's exact tests. Cox regression models were fitted using the coxph function in R "survival" package (v3.5-5). Feature selection within nested cross validation was conducted by LASSO using R package *glmnet* (v4.1-8). Multiple testing correction was conducted with the Benjamini-Hochberg correction method.

All analyses were conducted on publicly available cohorts (TCGA and Sato et al.). No new samples were generated for this study. Sample sizes reflect the number of patients available after applying the stated inclusion criteria.

## Results

### Clinical characteristics of cohorts

We performed extended clinical follow-up for metastatic events (median follow-up time of 23.2 months) in 192 ccRCC primary tumors profiled by TCGA[7] (Fig. 1, Supplementary Data 2 and Supplementary Fig. 3). We classified stage III primary tumors without subsequent metastasis during the entire follow-up as "indolent" (IN; $n = 68$) (Fig. 1a–c). Non-metastatic primary tumors that developed metastasis more than 100 days following diagnosis were classified as "metachronous metastatic" (MM; $n = 44$) (Fig. 1a–c). Finally, stage IV primary tumors or tumors developing metastasis within 100 days following diagnosis were classified as "synchronous metastatic" (SM; $n = 80$) (Fig. 1a–c). Consistent with previous reports[13], patients with SM tumors had significantly lower BMI as compared to other tumors (Kruskal-Wallis, $P = 0.0014$) (Supplementary Fig. 2), while no

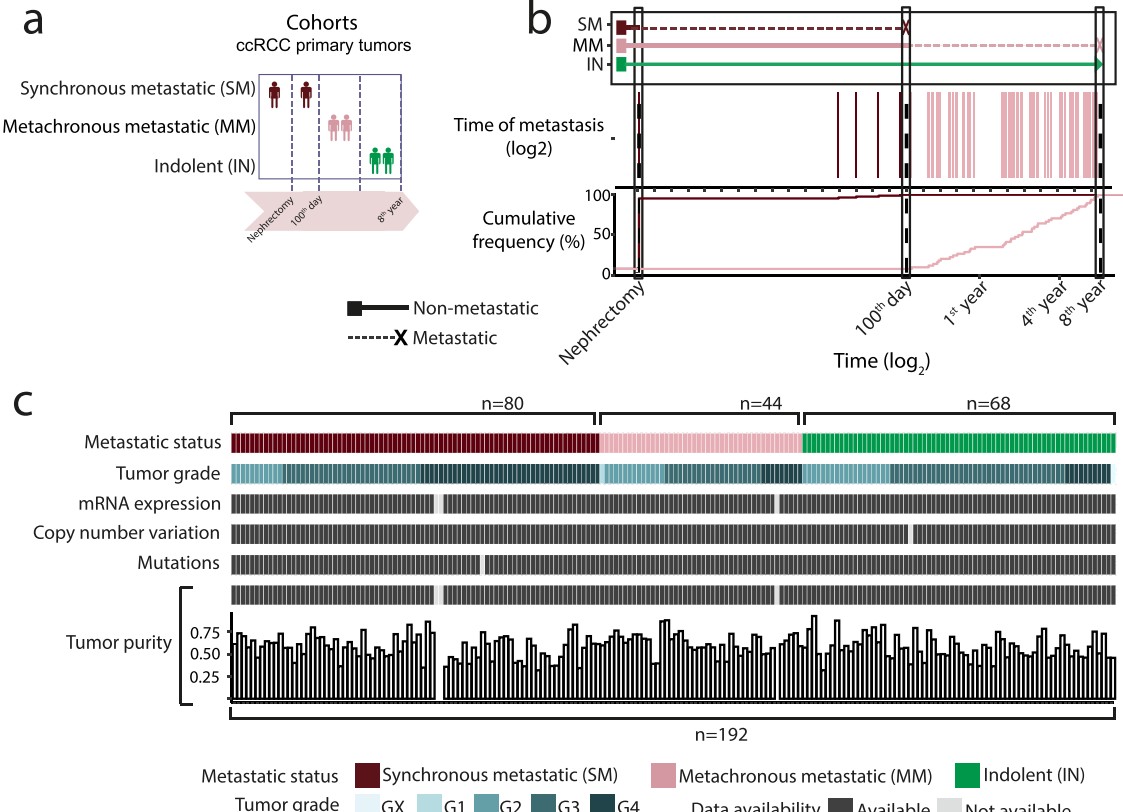

**Fig. 1 | Overview of ccRCC cohort. a** Schematic presentation of ccRCC cohorts. **b** Tumor samples grouped by metastatic progression status. Synchronous metastatic (SM) primary tumors were metastatic at presentation or <100 days following nephrectomy. Metachronous metastatic (MM) primary tumors developed delayed metastasis, >100 days following nephrectomy. Indolent (IN) primary tumors had no reported metastasis during the clinical follow-up time. **c** Available genomic and molecular data across the ccRCC cohort and metastatic progression status.

significant difference was found between IN and MM tumors (Wilcoxon rank-sum, $P = 0.3$). Similarly, SM tumors were larger (Kruskal-Wallis, $P = 6.4e-04$), with no difference between MM and IN tumors (Wilcoxon rank-sum, $P = 0.22$). Overall, these observations are consistent with previously reported features of metastatic ccRCC and support the notion that MM tumors are indistinguishable from IN tumors at the macroscopic level. To further characterize and validate the molecular findings of our analysis, we used an external cohort comprising 69 non-metastatic and 32 metastatic tumor samples with gene expression data[6] (Supplementary Fig. 3).

### Chromosome 1p33 loss is associated with metachronous metastasis

We first examined the genomic profiles in the three tumor subsets. Somatic mutation frequencies were not significantly associated with synchronous or metachronous metastasis in our cohort (Fig. 2a). Mutations in *BAP1* and *SETD2* have previously been linked to worse cancer-specific survival[35]. Indeed, our analysis also highlighted the increased mutation frequency of both genes in metastatic patients (Fig. 2a). *FREM1*, an extracellular matrix protein, showed enrichment of frameshift deletion and missense mutations in MM tumors ($n = 4$) as compared to IN tumors ($n = 0$) (Fig. 2a and Supplementary Fig. 4). However, this difference was not significant after multiple testing correction (Fisher's exact test, $P = 0.022$, $q$-value = 1).

Next, we examined the frequency of chromosomal CNA in the three groups. We identified CNAs at five regions associated with MM tumors (Wilcoxon rank-sum, $q$-value < 0.2) (Fig. 2b). These alterations comprised copy number gain at 19q12-13, 19p13, 2q14-36 and 2p12-16, and copy number loss across 1p31-36 (Fig. 2c and Supplementary Fig. 5a). We identified a strong positive correlation between CNA and gene expression levels in 1p31-36, which was not observed in the other four regions (Supplementary Fig. 5b). Similarly, genes located at the center of this region (1p33) showed

lower expression in MM compared to IN tumors (Wilcoxon rank-sum, $q$-value < 0.2) (Fig. 2d and Supplementary Fig. 6). A closer inspection of 1p33 revealed multiple altered genes (*NSUN4, FAAH, MKNK1, ATPAF1, CYP4A11, CYP4A22, STIL, CMPK1, TRABD2B*) with genomic loss and significantly down-regulated expression in MM compared to IN tumors (Wilcoxon rank-sum, $q$-value < 0.25) (Fig. 2e). While recurrent 1p36 copy number loss has been reported in multiple ccRCC cohorts[6–8], a previous report did not identify 1p36 loss as a selected and clonally expanded driver event in distant metastatic lesions[36].Taken together, these results suggest a relationship between chromosomal instability and metastatic seeding potential in localized ccRCC lesions, notably involving copy number loss and concomitant down-regulation of gene expression at or near 1p33-36.

### Depletion of regulatory T cells and M1 macrophages in MM tumors

To explore the role of TME immune cell infiltration as a risk factor for metastatic progression, we analyzed the immune cell type composition using the tumor RNA-seq profiles. We used three distinct methods for deconvolution of immune cell subsets from bulk tumor transcriptomic data: xCell[25], Consensus[TME][26] and Cibersort[27]. xCell identified strong depletion of T regulatory cells (Tregs) ($P = 2.99e-09$) (Fig. 3a) and M1 macrophages ($P = 6.6e-03$) in MM as compared to IN tumors. Similarly, Consensus[TME] confirmed the depletion of M1 macrophages and Tregs ($P = 0.1$ and $P = 0.022$) (Fig. 3a). Although not significant, depletion of these two cell types was also identified by Cibersort ($P = 0.40$ and $P = 0.30$) (Supplementary Fig. 7). We then compared the expression of individual Treg and M1 macrophages marker genes in MM and IN tumors (Supplementary Fig. 8 and 9). Among Treg gene markers, *MCM9, TULP4, STAM* and *IL10RA* showed significant down-regulation in MM tumors ($P < 0.01$) (Fig. 3b). Similarly, we identified significant down-regulation of multiple M1 macrophages marker genes

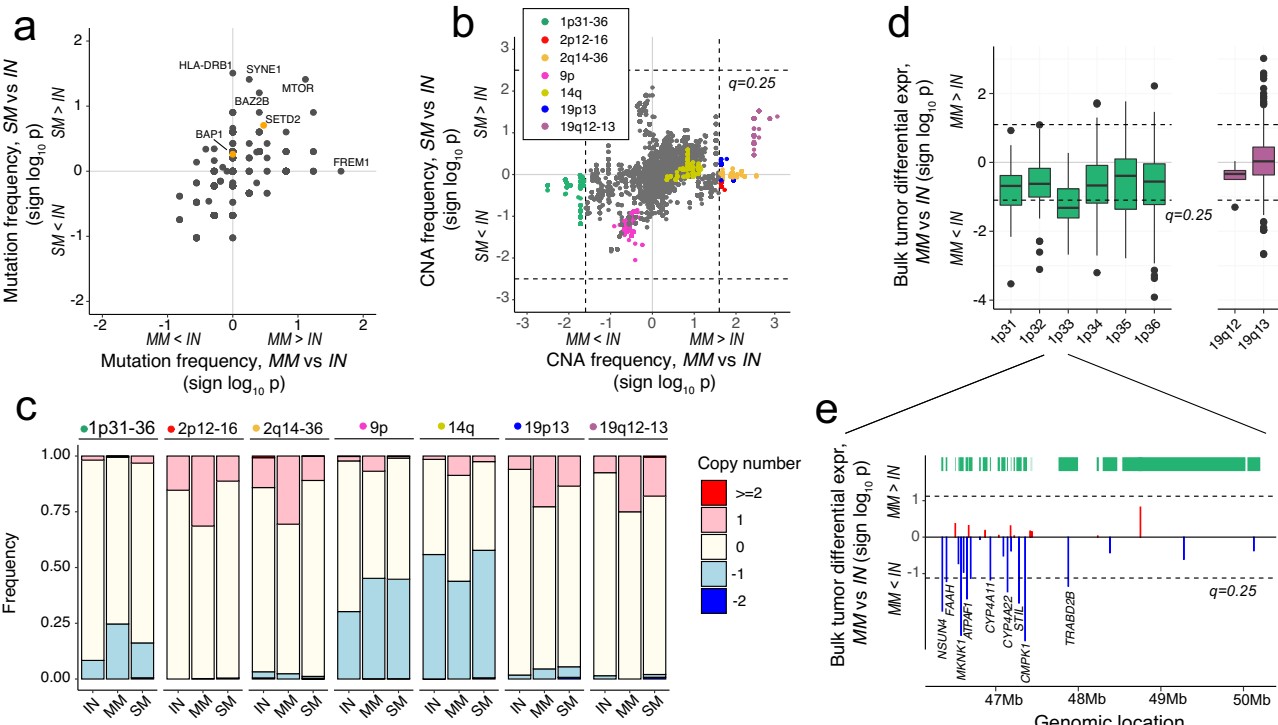

**Fig. 2 | Association of metastasis with mutations and copy number alterations. a** Comparison of gene mutation frequencies in metastatic (SM and MM) versus indolent (IN) tumors (two-tailed Fisher's exact test). No genes were significantly enriched in metastatic tumors after correcting for multiple testing. Known driver genes, *BAP1* and *SETD2*, are highlighted in orange color. **b** Comparison of copy number alteration frequencies in metastatic (SM and MM) versus indolent (IN) tumors (two-tailed Wilcoxon rank-sum test). Loci with significant ($q < 0.25$) differences between MM and IN cohorts are highlighted. Additionally, regions (9p and 14q) previously associated with distant metastasis are highlighted. **c** Copy number alteration frequencies of significant and highlighted regions across IN, MM and SM cohorts. **d** Comparison of bulk tumor expression of genes located in the 1p31-36 (CNA loss) and 19q12-13 (CNA gain) regions in MM ($n = 41$) and IN ($n = 62$) cohorts (two-tailed t-test). Exact p-values are provided in Supplementary Data 1. **e** Bulk tumor differential expression (sign $\log_{10}$ p, t-test) between MM and IN samples for individual genes located on 1p33. Significantly dysregulated genes are labeled ($q < 0.25$). Box plots in d display the median values with the interquartile range (lower and upper hinge) and ±1.5-fold the interquartile range from the first and third quartile (lower and upper whiskers). IN indolent cohort, MM metachronous metastasis cohort, SM synchronous metastasis cohort, CNA copy number alteration.

(*NDUFS2*, *FKBP15* and *SLC31A1)* in MM tumors ($P < 1e-03$) (Fig. 3c). Further investigation of these marker genes in SM tumors demonstrated consistent changes in gene expression for MM and SM tumors, but often with stronger expression changes in MM samples (Fig. 3b, c).

### Dysregulation of epithelial cell polarity associated with risk of metastatic progression

We conducted a systematic and unbiased analysis of gene expression differences in the three tumor subsets. A principal component analysis of bulk tumor transcriptomic data using the top 3000 variable genes could not stratify IN, MM and SM patients (Supplementary Fig. 10a–d). We then performed a comparative analysis of potential differences in the transcriptome profiles using a previously published tumor transcriptome deconvolution approach[20], which uses tumor purity to infer expression profiles of cancer and stromal cells inside the tumors (see Methods, Fig. 4a). Tumor purity was estimated using the genomic and transcriptomic profiles for each sample (see Methods). The samples showed a mean tumor purity of 49% (range 24–82%) with no significant differences between metastatic groups ($P = 0.067$).

We then compared the deconvoluted cancer cell gene expression profiles across the metastatic groups to identify putative gene expression signatures associated with the risk of metastatic progression. Relative to IN tumors, MM cancer cells demonstrated 87 down-regulated and 6 up-regulated genes ($q < 0.25$, Fig. 4b and Supplementary Data 3). *PATJ*, a tight junction and Crumbs cell polarity component, was the most significant down-regulated gene in MM tumors ($P = 2e-05$) (Fig. 4b). Expression of *PATJ* was also significantly down-regulated in cancer cells of SM samples ($P = 5.8e-04$) (Fig. 4c), suggesting a role for *PATJ* in metastatic progression.

To further explore the robustness of this association, we compared bulk tumor expression of *PATJ* in an independent cohort of metastatic and indolent ccRCC samples[6]. This analysis also confirmed significant down-regulation of *PATJ* in metastatic tumors ($P = 2.8e-03$) (Fig. 4c). Intriguingly, *PATJ* is also located on 1p31, displaying frequent copy number loss in MM tumors, indicating a convergence of genomic and transcriptomic alterations underlying risk of metastatic progression.

*TMSB10* was the most significantly up-regulated gene in cancer cells of MM tumors ($P = 3.4e-04$), with expression also up-regulated in SM ($P = 8.1e-03$) and metastatic tumors from Sato et al. ($P = 1.4e-03$) (Fig. 4c). *TMSB10* encodes for a member of the beta thymosin protein family which regulate cytoskeleton organization. Previous reports have also suggested a role for this protein in regulation of epithelial cell polarity in breast and ccRCC tumors[37,38]. Observing that multiple of the top dysregulated genes in MM tumors were involved in regulation of cell polarity, tight junction formation, and actin assembly, we examined the expression profiles of other known components of these processes[37,38]. This analysis revealed significant down-regulation of additional factors such as *AMOLT2*, *CGN* and *TJP2* (Supplementary Fig. 11a–c). Overall, these results highlight a putative role for dysregulation of tight junction and apicobasal cell polarity in cancer cells as a risk factor for the development of metachronous metastasis.

### Gene expression signatures of tumor stromal cells associated with metastatic risk

We next compared the deconvoluted stromal cell (comprising all non-malignant cells in the tumors) gene expression profiles across metastatic groups to identify tumor stroma gene expression signatures associated with

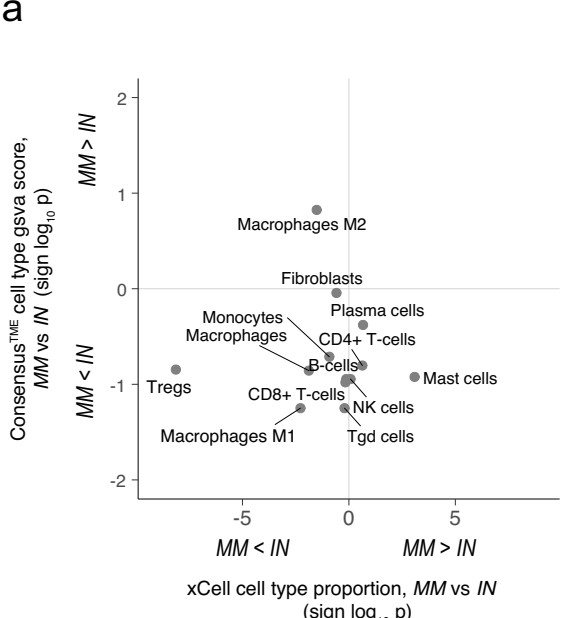

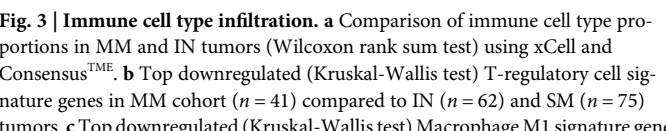

**Fig. 3 | Immune cell type infiltration. a** Comparison of immune cell type proportions in MM and IN tumors (Wilcoxon rank sum test) using xCell and Consensus[TME]. **b** Top downregulated (Kruskal-Wallis test) T-regulatory cell signature genes in MM cohort (n = 41) compared to IN (n = 62) and SM (n = 75) tumors. **c** Top downregulated (Kruskal-Wallis test) Macrophage M1 signature genes

in MM cohort (n = 41) compared to IN (n = 62) and SM (n = 75) tumors. Box plots in (**b**, **c**) refer to the median values with the interquartile range (lower and upper hinge) and ±1.5-fold the interquartile range from the first and third quartile (lower and upper whiskers). Tregs T regulatory cells, IN indolent cohort, MM metachronous metastasis cohort, SM synchronous metastasis cohort.

the risk of metastatic progression. We identified 719 down-regulated and 240 up-regulated genes in the stroma of MM as compared to IN tumors (q < 0.25, Supplementary Data 4). *LRRC19*, a pathogen-recognition receptor and inducer of pro-inflammatory cytokines, was strongly down-regulated in MM tumors (P = 1.0e-05). The expression of *LRRC19* was also down-regulated in stromal cells of SM samples (P = 1.0e-05) (Fig. 4c) and showed down-regulation in an independent cohort of metastatic tumors (P = 8.9e-02) (Fig. 4c). We also observed stromal down-regulation of *CNDP2*, encoding a nonspecific dipeptidase involved in histidine metabolism, in MM tumors (P = 1.0e-05) as well as the two cohorts of metastatic tumors (P = 1.7e-03; P = 3.4e-02) (Fig. 4c). Further analysis of public single-cell RNA-seq data from normal kidney tissue[39] showed that both *LRRC19* and *CNDP2* had the highest expression in proximal tubular cells (Supplementary Fig. 12). Furthermore, other top differentially expressed genes in the stroma (Fig. 4c) had the highest expression in either proximal tubular (*LRRC19*, *CNDP2*, *FJX1*), distal tubular (*TMEM230*), or collecting duct cells (*EVC*, *CDK2AP1*) (Supplementary Fig. 12). Taken together, these results indicate that many of the inferred stromal gene expression differences could be related to compositional differences of healthy kidney tissue cell types in MM and IN tumors

### Dysregulation of fatty acid metabolism associated with metachronous metastasis

To further explore the biological features associated with the MM tumor subset, we performed pathway-based enrichment analyses of genes with DE in cancer cells of MM and IN tumors (Fig. 4b). We observed significant down-regulation of genes associated with metabolic pathways (P = 1e-04, Fisher's exact test) and fatty acid degradation (P = 6e-04) (Fig. 5a). Several genes in the fatty acid degradation (FAD) pathway displayed significant down-regulation in MM as compared to IN samples (q < 0.1) (Fig. 5b). Among the top 10 down-regulated genes in the FAD pathway, 5 out of 10 genes also showed significant down-regulation in the two metastatic cohorts (P < 0.05, Fig. 5c). *CYP4A11*, an enzyme of the cytochrome P450 family involved in fatty acid omega oxidation, displayed very strong down-

regulation ( > 4-fold) across both, MM and SM, metastatic groups. Strikingly, *CYP4A11* is encoded at 1p33, displaying frequent copy number loss and concomitant gene expression down-regulation in MM tumors (Fig. 2e). Fatty acids (FAs), a source of energy and essential components of biomembranes, are degraded through three distinct pathways: α-, β- and ω-oxidation. We observed down-regulation of gene expression along all three different routes of fatty acid oxidation in MM as compared to IN tumors (Fig. 5d and methods). Overall, these results suggest that down-regulation of FAD, potentially leading to the accumulation of FAs and lipid droplets ('clear cell' phenotype) inside the cancer cells, could be a feature of cancer cells at risk of metastatic progression.

### Predicting risk of metachronous metastasis using a 5-gene signature

Pre-metastatic molecular features associated with the risk of developing metachronous metastasis could be incorporated into a model to predict risk of metastatic relapse. To build such a model, 94 patients (41 MM and 53 IN) were included, having at least 100 days of follow-up and available RNA-seq. The median age was 62.5 years (range: 34–88), with 64 males and 30 females.

We first performed a univariate Cox regression analysis on all putative genomic and molecular biomarker features identified in our study. Ten patients were excluded for the univariate cox regression analysis due to missing values (eight missing BIM, 2 missing CNV), leaving 84 patients. The individual predictive value of these features for metachronous metastasis risk was assessed (Fig. 6a). We selected the top five predictive gene expression features (*FKBP15, SLC31A1, CPT2, PATJ,* and *CALR*) to train a model (5G model) for predicting the risk of metachronous metastasis on all 94 patients without missing gene expression values. We used a cross-validation approach to evaluate the accuracy of our model on unseen data, dividing the dataset into training (80%) and test sets (20%). We fitted multivariate Cox regression models using the signature on the training data, evaluating the error on the withheld test set (cross-validation). This process was repeated 100 times to estimate the average accuracy (AUC) of each model on the unseen test data. Previous studies have also proposed distinct

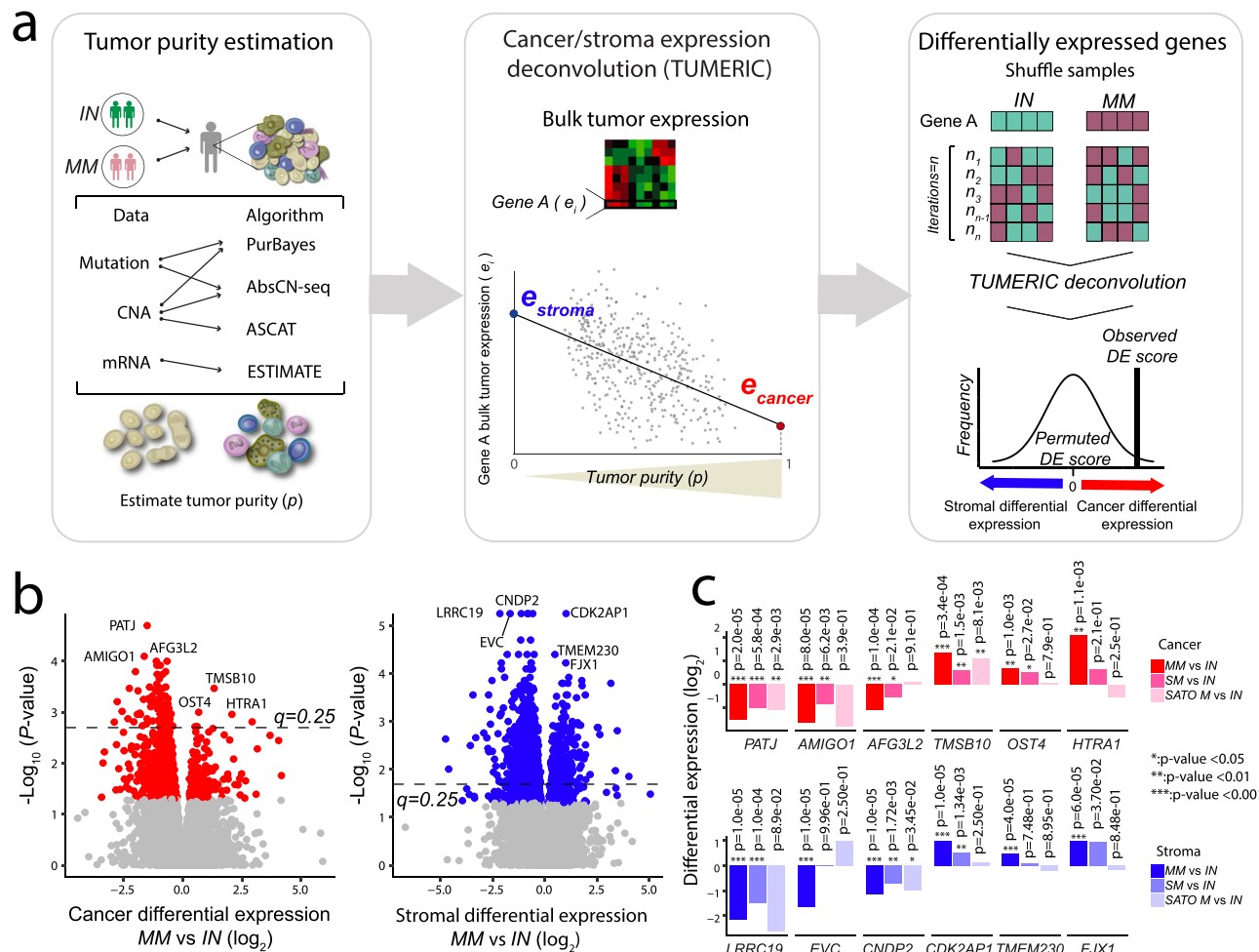

**Fig. 4 | Cancer vs. stroma transcriptome deconvolution. a** Tumor purity of each sample is jointly estimated from DNA and RNA data using a consensus approach. Bulk tumor gene expression is then deconvoluted into cancer and stroma cell components. Genes with differential cancer or stromal cell expression across groups (IN and MM) are identified using a sample permutation approach. **b** Cancer and stromal differential expression comparing MM vs. IN cohorts. Blue and red points represent significant genes based on empirical *P*-values (0.05) from a two-sided permutation test; the dashed horizontal line indicates the false discovery rate (FDR) threshold (*q* = 0.25). **c** The top differentially expressed genes are compared across groups, including an independent cohort of metastatic ccRCC tumors (SATO M, Sato et al. 2013). Empirical *p*-values are derived using a two-sided non-parametric permutation test. Differential expression relative to the IN group is displayed; ***: *p* < 0.001; **: *p* < 0.01; *: *p* < 0.05 IN indolent cohort, MM metachronous metastasis cohort, SM synchronous metastasis cohort, SATO M SATO et al. metastatic cohort.

gene signatures predictive of ccRCC disease recurrence[14,15,18] and mortality within 5 years of radical nephrectomy[16]. We compared the predictive accuracy of our two models with these existing signatures in stratifying MM from IN patients in our cohort. The 5G signature model demonstrated the best accuracy (73.7% AUC), with a specificity of 38.5% evaluated at 87.8% sensitivity. The existing Rini et al.[15] (55.6% AUC, 16.24% specificity at 87.8% sensitivity), Morgan et al.[16] (45.23% AUC, 10.67% specificity at 87.8% sensitivity), MSKI[18] (56.32% AUC, 19.5% specificity at 87.8% sensitivity), clearcode34/ccB[14] (60.92% AUC, 25.03% specificity at 87.8% sensitivity), and Alchahin et al.[19] (61.25% AUC, 25.66% specificity at 87.8% sensitivity) signatures demonstrated overall lower accuracy in predicting metachronous metastasis risk (Fig. 6b). We further evaluated the association of the 5G model scores with disease-free survival on 92 patients, excluding two patients with missing vital status information. Splitting patients into tertiles based on the 5G score, the model demonstrated a significant difference in outcomes for patients with low and medium/high scores (Fig. 6c, *p* = 0.0088, log-rank test). Overall, these results suggest that the 5G model could be used to improve prediction of metastatic risk and stratify ccRCC patients into low and high-risk groups following curative surgery.

To further evaluate the robustness of our model training approach, we performed repeated nested cross-validation with independent feature-selection using cox regression with LASSO within each fold (Supplementary Fig. 13). This approach yielded comparable model performance (70.6% AUC, s.d. 0.097) to the 5G model (73.7%, s.d. 0.092). We also performed an analysis of the gene selection frequency across the nested CV runs, showing that all five genes in the 5G signature were selected in the top eight in this analysis (Supplementary Fig. 14). Expression correlations among the top-selected genes likely contribute to variability between the different CV runs (Supplementary Fig. 15). Overall, these analyses underscore that all five genes carry robust biological signals for predicting the risk of metachronous metastasis.

Finally, we validated the robustness of the 5G signature in an external dataset, we trained a cox model to predict overall survival using the 5G signature and evaluated its performance on the Sato dataset[6]. Patients were divided into tertiles based on their 5G scores, yielding significant differences in outcomes for patients with predicted low, medium, and high-risk scores (Supplementary Fig. 16, *p* = 0.0012, log-rank test).

## Discussion

Our study aimed to identify cancer and stromal cell molecular features as putative biomarkers of localized ccRCC primary tumors at risk of metastatic progression. We investigated the genomic and transcriptomic profiles of

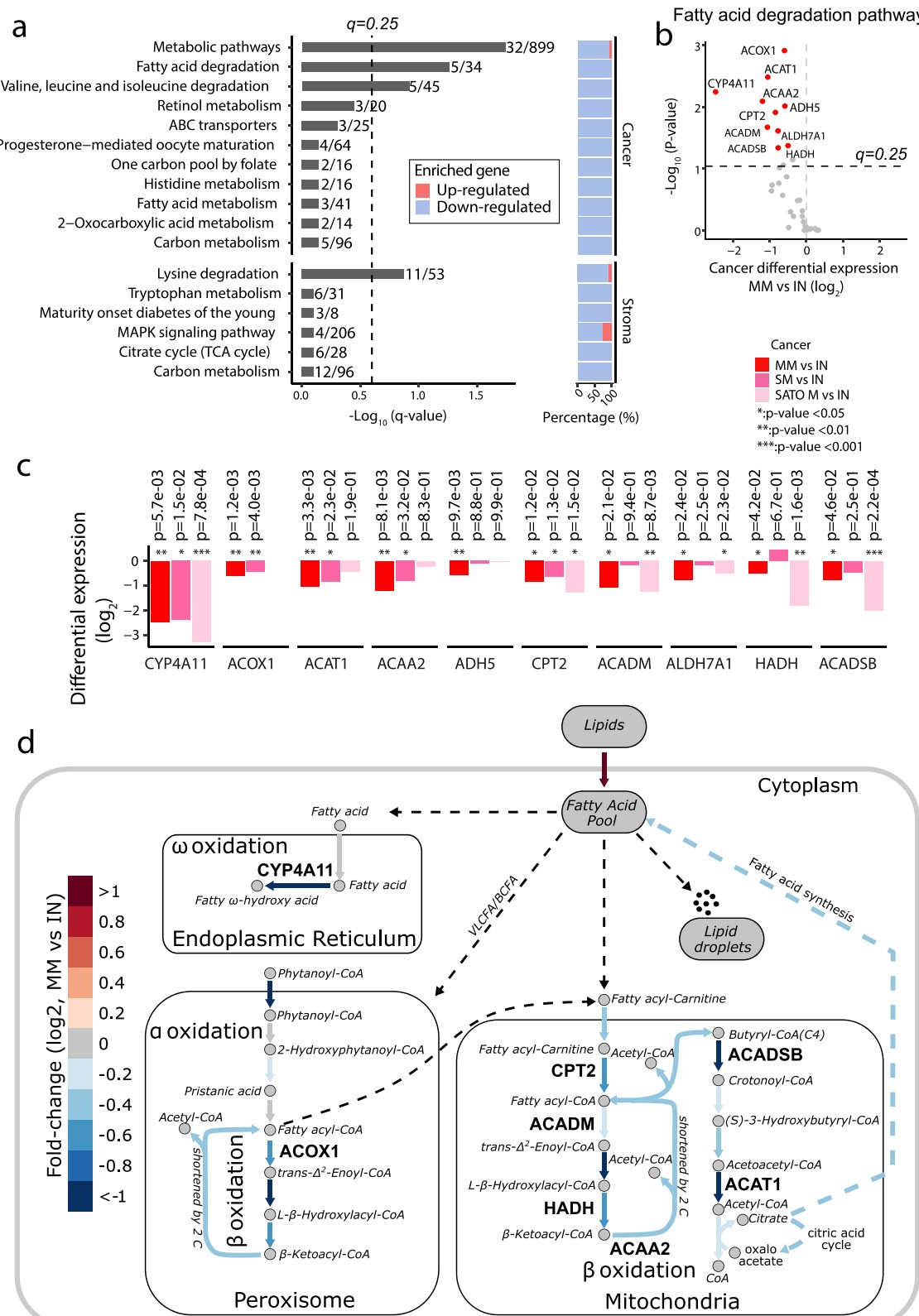

**Fig. 5 | Fatty acid metabolism associated with risk of metachronous metastasis.**
**a** Pathway enrichment of cancer and stromal differentially expressed genes in MM vs IN cohorts (Fig. 4b, P < 0.05). The proportion of up and down-regulated genes is indicated for each pathway. P-values are derived by Wilcoxon rank-sum test and adjusted to FDR *q*-values using the Benjamini-Hochberg method. **b** Inferred cancer-cell expression differences (MM vs IN) of genes involved in the fatty acid degradation (FAD) pathway. Points represent empirical P-values obtained from a two-sided non-parametric permutation test; genes above the dashed horizontal line are significant based on FDR-adjusted q-values (*q* = 0.25). **c** Differential expression of significant FAD genes meeting the significance threshold (q < 0.1) across groups and external SATO-M cohort. Empirical P-values were derived using a two-sided non-parametric permutation test; significance is defined by FDR-adjusted q-values. **d** Map of FAD metabolic reactions. Metabolites and reactions are indicated with circles and arrows, respectively. Reactions are color-coded according to cancer-cell expression changes (MM vs. IN) of genes/enzymes in the given reaction. Genes highlighted in panel c are labeled on the map.

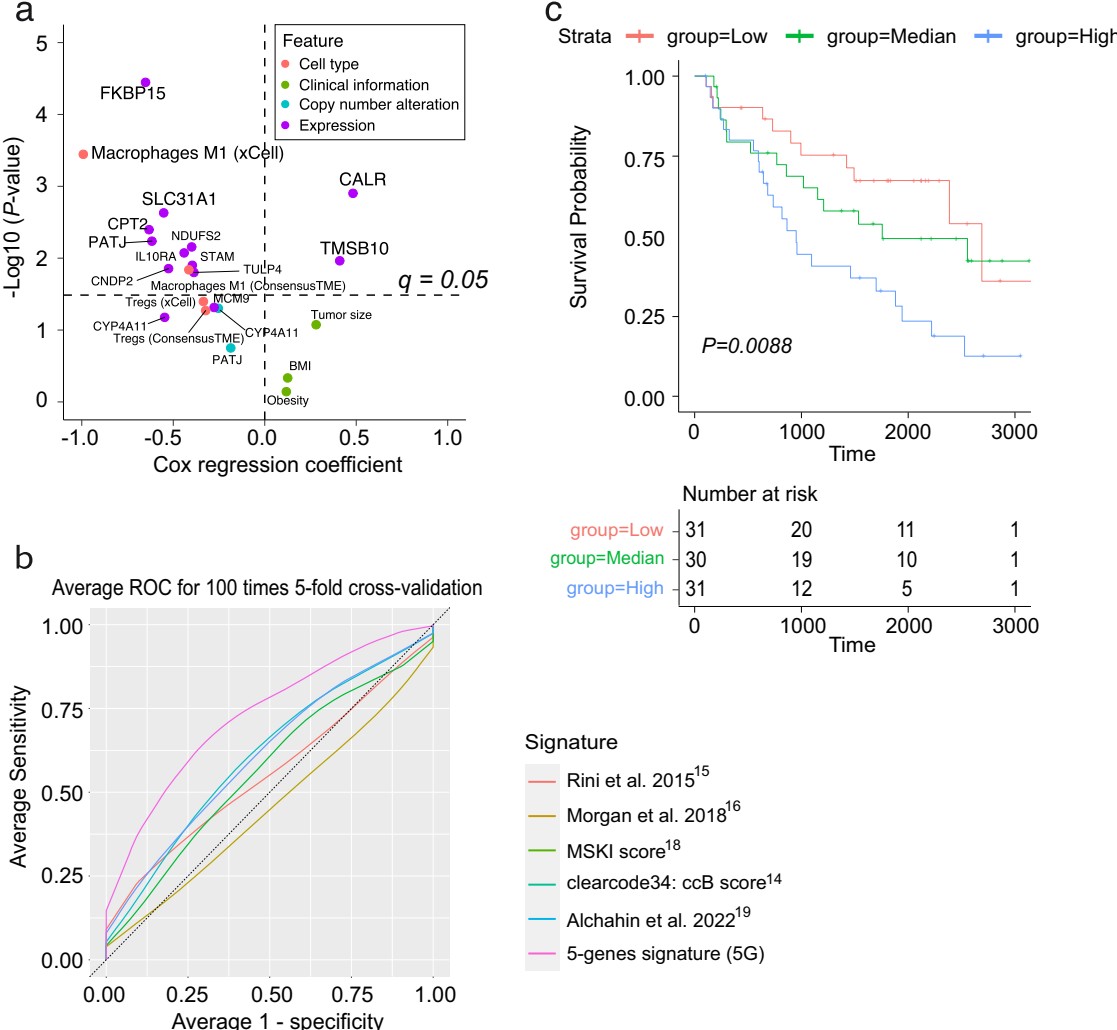

**Fig. 6 | Genomic and molecular features predictive of metachronous metastasis risk. a** Cox regression analysis of individual candidate features associated with the risk of metachronous metastasis in this study. Points represent P-values obtained from a two-sided Wald test; features above the dashed horizontal line are significant based on FDR-adjusted q-values ($q = 0.05$). **b** Average ROC for 100 times 5-fold cross-validation for prediction of metachronous metastasis risk; comparing the performance of the 5-gene (5G) signature including FKBP15, SLC31A1, CPT2, PATJ and CALR (purple) versus the ccRCC metastasis gene signatures previously published by Rini et al. 2015[15] (red), Morgan et al. 2018[16] (olive), MSKI score[18] (green), clearcode34: ccB score[14] (teal) and Alchahin et al. 2022[19] (blue). **c** Kaplan-Meier plot of disease-free survival (log-rank test) with the number of patients displayed for each group shown in the table below the plot.

192 ccRCC primary tumors. In our study, these patients were analyzed using extended clinical follow-up beyond the standard TCGA clinical annotation. This allowed us to define subsets of high-confidence indolent tumors as well as tumors that at presentation appeared localized but later relapsed with metachronous metastasis (MM). We identified genomic aberrations as well as cancer and stroma cell gene expression signatures that characterize these primary tumors at risk of developing metastasis. Strikingly, these genomic and expression signatures converged at the 1p31-36 locus, displaying both frequent copy number loss and gene expression down-regulation in patients developing MM. Finally, these insights enabled us to develop a compact prognostic model to predict the risk of metastatic progression in ccRCC patients with localized disease.

Previous studies have reported altered frequencies of *BAP1* and *SETD2* driver mutations when comparing primary ccRCC tumors and metastatic tumors[8]. Our study also identified an increased frequency of these mutations in metachronous metastatic tumors. However, this association was not statistically significant in our cohort, possibly due to a lower sample size and the restriction to stage III patients in this study. Instead, our analysis revealed that primary tumors display an increased probability of metastatic progression when they harbor specific genomic copy number aberrations. Specifically, we demonstrated that loss of chromosome 1p and gains of 2p, 2q, 19p and 19q

were associated with the risk of metachronous metastasis. Consistent with previous studies of metastatic ccRCC lesions[8], we also found that 2q14.3 gain and 1p36.11 loss are more prevalent in tumors that later progress with metastasis. Moreover, our integrative analysis found that the 1p31-36 region (especially 1p33) also harbored genes, including *PATJ* and *CYP4A11*, that were significantly down-regulated in tumors that progressed with metastasis.

Using bulk tumor gene expression deconvolution, we observed significant down-regulation of *PATJ* expression in cancer cells of tumors that progressed with metachronous metastasis. *PATJ* encodes for a component of the Crumbs cell polarity complex and is involved in the formation of tight junctions as well as the apicobasal cell polarity[40,41]. Further analysis of factors involved in the regulation of tight junctions and cell polarity (Crumbs, Par, Scribble complexes) identified additional genes (*AMOLT2*, *CGN* and *TJP2*) with down-regulation in tumors that developed metastasis. Additionally, the top up-regulated gene in cancer cells of MM tumors was *TMSB10*, which has been linked to the regulation of cell polarity and metastatic invasion in breast cancer and ccRCC[37,42]. Consistent with existing knowledge on the role of these biological processes in metastatic seeding and progression[43-45], our data suggest that primary ccRCC cancer cells with dysregulated epithelial cell polarity and tight junction formation exhibit an elevated risk of progressing with metastasis.

Our analysis also highlighted a link between metabolic reprogramming and the risk of metastatic progression. A hallmark of ccRCC is the accumulation of FAs as lipid droplets inside cancer cells, the 'clear-cell' phenotype[36]. FAs are a source of energy and constitute essential components of bio-membranes. FAs are degraded through three distinct pathways including α-, β- and ω-oxidation. Here we report the down-regulation of gene expression along all three different pathways of fatty acid oxidation in tumors that later progressed with metastasis. In the ω-oxidation pathway, *CYP4A11* displayed very strong down-regulation ( > 4-fold) across all metastatic cohorts in our analysis. Intriguingly, *CYP4A11* is also encoded on the 1p33 chromosome locus displaying frequent copy number loss and gene expression down-regulation in tumors at risk of metastatic progression. We also observed strong down-regulation of Palmitoyltransferase-2 (*CPT2*), which is crucial for FA β-oxidation through the carnitine shuttle system into the mitochondrial matrix. FA β-oxidation has been demonstrated to be suppressed by hypoxia-inducible factors (HIFs) and is possibly essential for ccRCC tumorigenesis[46]. Intriguingly, our study indicates that tumors with reduced capacity for FA oxidation also exhibit a more aggressive phenotype with an elevated risk of metastatic progression. These findings should also be interpreted in the context that ccRCC tumors developing in an obesogenic environment may generally be more indolent[5].

We also identified a potential role for the TME in tumors at risk for metachronous metastasis. Using multiple immune cell deconvolution algorithms, we identified significant depletion of T-regulatory (Tregs) and M1 macrophage cell subsets in tumors at risk of metastatic progression. Consistent with these results, elevated Treg abundance in the TME has been observed in patients with favorable outcomes across diverse tumor types[47,48]. Moreover, M1 macrophages are generally considered 'good' macrophages associated with increased anti-tumor inflammation[49]. Notably, a recent study suggested that intratumoral myeloid inflammation is associated with metastatic progression[18]. In line with our result, this study highlighted the enrichment of M1 macrophages in control mice without metastatic progression. In contrast, it has been suggested that M2 macrophages are associated with exhausted CD8 T-cells, advanced tumor stages, and metastatic progression[18,50]. Altogether, our results are concordant with these previous observations that metastatic progression of ccRCC is associated with a unique myeloid inflammatory state depleted from M1 macrophages. Our immune cell type deconvolution was based on the default LM22 signature provided by Cibersort. It is conceivable that a ccRCC-specific scRNA-derived reference signature could provide higher-resolution deconvolution of immune and stromal subtypes in these tumors. However, no standardized and validated ccRCC signature has been published, representing an important future direction for delineating the biological factors that underlie metachronous metastasis.

Finally, we developed and evaluated a compact predictive model based on the molecular features associated with patient tumors developing metachronous metastasis. The 5G model, incorporating *FKBP15, SLC31A1, CPT2, PATJ*, and *CALR* expression, demonstrated improved accuracy in predicting risk of metastatic progression as compared to existing gene signatures proposed for the prediction of disease recurrence and mortality in ccRCC patients[14–16,18,19]. We validated the risk stratification performance in an external cohort[6]; however, additional validation needs to be conducted to further evaluate the robustness of this model. The potential clinical utility for identifying patients at risk of developing metachronous metastasis must also be evaluated in combination with existing clinical risk models for ccRCC[46].

In summary, we have identified convergent genomic and transcriptomic features associated with the risk of developing metachronous metastasis in ccRCC. Intriguingly, our study highlights chromosome 1p31-36 copy number loss, potentially driving dysregulation of both apicobasal cell polarity and fatty acid metabolism, as a key risk factor of metastatic seeding and progression in ccRCC. Our study also highlights a putative biomarker signature for the stratification of primary ccRCC tumors at risk of metastatic progression. Overall, to the best of our knowledge, these results provide new insights into the biology of metastatic ccRCC progression as well as potential new avenues for clinical management and risk stratification of early-stage ccRCC patients.

## Inclusion & ethics statement

This study aligns with the "Inclusion & ethical" guidelines by *Communications Medicine*.

## Data availability

All external datasets analyzed in this study are publicly available. The related source repositories for TCGA and Sato dataset are provided in Supplementary Table 1. The source data to generate Figs. 2–6 are in Supplementary Data 1 and the source data for Fig. 1 is provided in Supplementary Data 2.

## Code availability

Codes that support the findings of this study are publicly available in Zenodo (https://zenodo.org/records/17233343)[51].

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

## Acknowledgements

**MMN** was supported by the A*STAR Research Attachment Program (ARAP) PhD scholarship. The results shown here are based upon data generated by the TCGA Research Network: https://www.cancer.gov/tcga. We are grateful to the patients who participated in this project.

## Author contributions

M.M.N.: Conceptualization, Formal analysis, Methodology, Investigation, Visualization, Writing - Original Draft. M.P.: Formal analysis, Visualization, Investigation, Writing - Review & Editing. N.R.: Formal analysis, Visualization, Investigation, Writing - Review & Editing. S.K.: Investigation. U.G.: Formal analysis. R.G.D.: Data curation, Resources. R.M.: Data curation, Resources. A.A.H.: Conceptualization, Data curation, Resources, Investigation, Writing - Review & Editing. A.J.S.: Conceptualization, Data curation, Methodology, Formal analysis, Investigation, Supervision, Project administration, Writing - Original Draft.

## Competing interests

The authors declares no competing interests.
