## [Transparent Peer Review file · Communications Medicine]

Convergent genomic and molecular features predict risk of metachronous metastasis in clear cell renal cell carcinoma

Corresponding Author: Dr Anders Skanderup

Version 0:

Reviewer comments:

Reviewer #1

(Remarks to the Author)

Naeini et al. present an interesting study that takes advantage of extended clinical follow-up of a large cohort of genetically and transcriptionally characterised clear cell renal cell carcinoma patients from the TCGA study, as well as uses an independent cohort (Sato et al study) to validate findings. The study classifies ccRCC tumours into those that had metastasised at the time of nephrectomy (synchronous), within 100 days of nephrectomy (metachronous) or that did not metastasise within the follow-up period (indolent). Numerous state of the art bioinformatic approaches were employed to identify molecular differences between the cohorts of patients, including chromosome copy number alterations, deconvolution of tumour and stromal gene expression profiles, deconvolution of immune composition of the tumor immune microenvironment. These analyses collectively led to the development of a new 5 gene signature that outperforms all previously proposed signatures in terms of predicting metastatic spread. This signature may find potential clinical use to identify those patients that would benefit from additional therapy or closer monitoring. The findings also identify cell polarity and fatty acid oxidation as cellular processes that correlate with metastatic propensity. These observations will likely drive further research into their functions in the context of ccRCC biology and may potentially also identify future therapeutically targetable features of tumour cells with high metastatic capacity. The manuscript is very clearly written and the figures are presented at the highest standards. The abstract reflects the data that is presented in the paper and the discussion section nicely describes the conclusions as well as potential limitations of the current study. Overall this study makes an important scientific and clinical contribution to the field of ccRCC.

Reviewer #2

(Remarks to the Author)

I think this is an interesting contributions that has a potential to improve our understanding and prediction of metastatic ccRCCs. However, two major issues need to be addressed/resolved, before I can recommend it for publication.

The first deals with the optimization and evaluation of the proposed 5-gene predictive signature. From the brief description in the methods section, one is left with the impression that all tumors were used to select the most informative genes. Such selected features (top 5 genes) are then used to train and assess a model using 5-fold cross-validation. If there was a strict separation of training and validation sets, which is paramount for honest assessment of generalization, each training subsets should be used to select features without using the corresponding validation folds. This would have likely resulted in selecting different top genes in each run of cross-validation. Thus, most likely the most predictive features were selected using all data, rendering the estimates of accuracy on the "validation" sets invalid. Strict separation here is required to obtain realistic estimates of accuracy on new data. On a more technical note, given the repeated nature of cross-validation runs, one can obtain not only average accuracy metrics but also their variance (standard deviation), which should be reported as well (only average values are reported on page 9). Without those error bars it is impossible to say if the proposed new signature outperforms previous approaches.

The second issue deals with deconvolution. The deconvolution approaches used here, such as Cibersort, relied on predefined libraries of reference cell types and their expression signatures. While this can provide estimates of the immune system cell infiltration or fraction of transformed vs. stroma cells, it lacks the resolution of single cell studies. It also misses an opportunity to use Cibersort, Music or other methods in conjunction with a targeted cell type library derived from public domain single cell RNA-seq studies that largely mapped clonal heterogeneity and tumor microenvironments. These studies,

taken together provide an excellent resource to build such a reference library for the deconvolution of ccRCC bulk RNA-seq data and provide an opportunity to identify clonal subpopulations and their microenvironment that contribute to the risk of metastasis.

Version 1:

Reviewer comments:

Reviewer #1

(Remarks to the Author)

Reviewer #3

(Remarks to the Author)

I co-reviewed this manuscript with one of the reviewers who provided the listed reports. This is part of the Communications Medicine initiative to facilitate training in peer review and to provide appropriate recognition for Early Career Researchers who co-review manuscripts.

Reviewer #4

(Remarks to the Author)

The authors provided detailed responses for the Reviewer comments. Congrats.

Responses to reviewers

We thank the reviewers for their helpful suggestions and comments. We have responded to each of the comments from reviewers in our rebuttal document. When changes have been made to the manuscript, we have used track changes and provided both a tracked change and a clean version of the revised manuscript.

REVIEWER #1

1. Naeini et al. present an interesting study that takes advantage of extended clinical follow-up of a large cohort of genetically and transcriptionally characterised clear cell renal cell carcinoma patients from the TCGA study, as well as uses an independent cohort (Sato et al study) to validate findings. The study classifies ccRCC tumours into those that had metastasised at the time of nephrectomy (synchronous), within 100 days of nephrectomy (metachronous) or that did not metastasise within the follow-up period (indolent). Numerous state of the art bioinformatic approaches were employed to identify molecular differences between the cohorts of patients, including chromosome copy number alterations, deconvolution of tumour and stromal gene expression profiles, deconvolution of immune composition of the tumor immune microenvironment. These analyses collectively led to the development of a new 5 gene signature that outperforms all previously proposed signatures in terms of predicting metastatic spread. This signature may find potential clinical use to identify those patients that would benefit from additional therapy or closer monitoring. The findings also identify cell polarity and fatty acid oxidation as cellular processes that correlate with metastatic propensity. These observations will likely drive further research into their functions in the context of ccRCC biology and may potentially also identify future therapeutically targetable features of tumour cells with high metastatic capacity. The manuscript is very clearly written and the figures are presented at the highest standards. The abstract reflects the data that is presented in the paper and the discussion section nicely describes the conclusions as well as potential limitations of the current study. Overall this study makes an important scientific and clinical contribution to the field of ccRCC.

RESPONSE: We thank the reviewer's supportive feedback.

REVIEWER #2

I think this is an interesting contributions that has a potential to improve our understanding and prediction of metastatic ccRCCs.

RESPONSE: We thank the reviewer for appreciating the potential of our study.

However, two major issues need to be addressed/resolved, before I can recommend it for publication.

The first deals with the optimization and evaluation of the proposed 5-gene predictive signature. From the brief description in the methods section, one is left with the impression that all tumors were used to select the most informative genes. Such selected features (top 5 genes) are then used to train and assess a model using 5-fold cross-validation. If there was a strict separation of training and validation sets, which is paramount for honest assessment of generalization, each training subsets should be used to select features without using the corresponding validation folds. This would have likely resulted in selecting different top genes in each run of cross-validation. Thus, most likely the most predictive features were selected using all data, rendering the estimates of accuracy on the "validation" sets invalid. Strict separation here is required to obtain realistic estimates of accuracy on new data. On a more technical note, given the repeated nature of cross-validation runs, one can obtain not only average accuracy metrics but also their variance (standard deviation), which should be reported as well (only average values are reported on page 9). Without those error bars it is impossible to say if the proposed new signature outperforms previous approaches.

RESPONSE: We thank the reviewer for sharing this important point. In our original submission, we reported average accuracy values (AUC) from repeated 5-fold cross-validation (100 runs). To address the reviewer's concern, we now provide standard deviations alongside mean AUCs, thereby quantifying the variance of model performance across runs. **Supplementary Figure 13** shows that the 5-gene (5G) model achieved an average AUC of 73.7% (s.d. 0.092), outperforming previously published signatures such as Rini et al. (55.6%, s.d. 0.117) and clearcode34/ccB (60.9%, s.d. 0.108).

To further address the reviewer's point, we performed a new nested cross-validation (nested CV) analysis to enforce strict separation of training and validation folds for feature selection. In this scheme, feature selection (via LASSO Cox regression) was carried out independently within each training fold, and model evaluation was performed on the corresponding held-out validation fold. This procedure was repeated 100 times using the same data splits for comparability. **Supplementary 13** also illustrates that the nested CV model achieved an average AUC of 70.6% (s.d. 0.097), which is comparable to the original 5G model (73.3%) and higher than existing signatures (55.6%-60.9%).

Notably, **Supplementary Figure 14** shows that three of the five genes in the 5G signature (*FKBP15*, *CPT2*, *CALR*) were also among the five most frequently selected across the nested CV runs. The remaining two genes (*PATJ* and *SLC31A1*) were selected in the top eight. These two genes showed strong correlations with other genes (*PATJ*: *CPT2* $r=0.68$, *TSMSB10* $r=-0.66$; *SLC31A1*: *FKBP15* $r=0.61$, *STAM* $r=0.65$; **Supplementary Figure 15**), explaining how feature selection could vary slightly between cross-validation iterations.

In summary, our results confirm that the 5G signature is robust and may provide improved risk prediction when compared to existing published signatures.

Supplementary Figure 13: Boxplot of Area Under the Curve (AUC) of 100 times 5-fold cross-validation for prediction of metachronous metastasis risk. '5-genes signatures' model: the top-5 features were selected based on the full training set, and AUC subsequently determined using 100 iterations of five-fold cross-validation. 'Nested 5-fold CV': same approach, but with model features repeatedly selected within each cross-validation training fold.

Supplementary Figure 14: Frequency of gene selection across 100 repeated nested cross-validation runs.

Supplementary Figure 15: Correlation matrix of the genes selected during nested cross-validation (Suppl. Fig. 14), with gene locations (cytobands) indicated.

CHANGES TO MANUSCRIPT:

- We added a new **Supplementary Figure 13** showing the distribution of AUC values across 100 repeated 5-fold nested cross-validation runs.
- We included **Supplementary Figure 14** showing the frequency of gene selection across repeated nested CV runs, demonstrating that three of the five 5G genes (*FKBP15*, *CPT2*, *CALR*) were consistently among the top-5 most frequently selected; the remaining two genes (*PATJ* and *SLC31A1*) were selected in the top eight.
- We included **Supplementary Figure 15** showing the pairwise correlation structure of the 13 genes selected during the nested cross-validation.
- We updated the “**Result**” section to reflect his new analyses:

*“To further evaluate the robustness of our model training approach, we performed repeated nested cross-validation with independent feature-selection within each fold (**Supplementary Figure 13**). This approach yielded comparable model performance (70.6% AUC, s.d. 0.097) to the 5G model (73.7%, s.d. 0.092). We also performed an analysis of the gene selection frequency across the nested CV runs, showing that all five genes in the 5G signature were selected in the top eight in this analysis (**Supplementary Figure 14**). Expression correlations among the top-selected genes likely contribute to variability between the different CV runs (**Supplementary Figure 15**). Overall,*

these analyses underscore that all five genes carry robust biological signals for predicting the risk of metachronous metastasis.”

The second issue deals with deconvolution. The deconvolution approaches used here, such as Cibersort, relied on predefined libraries of reference cell types and their expression signatures. While this can provide estimates of the immune system cell infiltration or fraction of transformed vs. stroma cells, it lacks the resolution of single cell studies. It also misses an opportunity to use Cibersort, Music or other methods in conjunction with a targeted cell type library derived from public domain single cell RNA-seq studies that largely mapped clonal heterogeneity and tumor microenvironments. These studies, taken together provide an excellent resource to build such a reference library for the deconvolution of ccRCC bulk RNA-seq data and provide an opportunity to identify clonal subpopulations and their microenvironment that contribute to the risk of metastasis.

RESPONSE: We agree that a ccRCC-specific scRNA-based reference would likely provide higher deconvolution resolution than generic reference matrices such as LM22. LM22, developed from gene expression profiles of purified leukocyte subsets, provides broad immune categories (e.g., M1 vs. M2 macrophages). By contrast, a ccRCC-specific reference derived from single-cell data could capture the finer diversity of tumour-infiltrating immune subtypes, stromal populations, and malignant epithelial states observed in the ccRCC tumour microenvironment.

However, to the best of our knowledge, no standardized ccRCC scRNA-derived reference signature currently exists. Furthermore, the available single-cell datasets (e.g., Krishna *et al.*, *Cancer Cell* 2021) include only a handful of patients (n=6, comprising 29 multiregional samples), with just one metastatic case and no systematic clinical follow-up. While these data are valuable for describing tumour microenvironmental biology, they may not be sufficient to construct a robust, generalizable reference panel for prognostic validation in bulk ccRCC tumor data.

In light of these limitations, we have retained our bulk deconvolution approach (using the default LM22 Cibersort signature). In the revised manuscript, we now explicitly acknowledge that a ccRCC scRNA-derived reference could add further biological granularity, emphasizing that while such signatures are not yet available, they represent an important future direction for the field.

CHANGES TO MANUSCRIPT: We added a sentence in the “**Discussion**” to highlight this aspect:

“Our immune cell type deconvolution was based on the default LM22 signature provided by Cibersort. It is conceivable that a ccRCC-specific scRNA-derived reference signature could provide higher-resolution deconvolution of immune and stromal subtypes in these tumors. However, no standardized and validated ccRCC signature has been published, representing an important future direction for delineating the biological factors that underlie metachronous metastasis.”